# Prevalence, Diversity, and Risk Factors for Cervical HPV Infection in Women Screened for Cervical Cancer in Belém, Pará, Northern Brazil

**DOI:** 10.3390/pathogens11090960

**Published:** 2022-08-24

**Authors:** Jacqueline Cortinhas Monteiro, Mihoko Yamamoto Tsutsumi, Deivid Oliveira de Carvalho, Elenice do Carmo da Silva Costa, Rosimar Neris Martins Feitosa, Rogério Valois Laurentino, Ricardo Roberto de Souza Fonseca, Rodrigo Vellasco Duarte Silvestre, Aldemir Branco Oliveira-Filho, Luiz Fernando Almeida Machado

**Affiliations:** 1Biology of Infectious and Parasitic Agents Post-Graduate Program, Federal University of Pará, Belem 66075-110, PA, Brazil; 2Virology Laboratory, Institute of Biological Sciences, Federal University of Pará, Belem 66075-110, PA, Brazil; 3Laboratory of Cytopathology, Institute of Biological Sciences, Federal University of Pará, Belem 66075-110, PA, Brazil; 4Evandro Chagas Institute, Health Ministry of Brazil, Ananindeua 67030-000, PA, Brazil; 5Study and Research Group on Vulnerable Populations, Institute for Coastal Studies, Federal University of Pará, Bragança 68600-000, PA, Brazil

**Keywords:** epidemiology, viruses, viral genetics, infectious diseases, HPV

## Abstract

Background: Human papillomavirus (HPV) is the most common viral sexually transmitted infection of the reproductive tract, and cervical cancer is the most common HPV-related disease. This study estimated the prevalence, diversity of HPV genotypes, and associated risk factors in women screened for cervical cancer in northern Brazil. Methods: The cross-sectional study was conducted in Belém, Pará, in the Amazon region of Brazil, and it included 162 women who were spontaneously undergoing a Pap-smear routine. Epidemiological, sexual, and health-related information was collected by interviews, and cervical samples were collected for cytological examination and HPV-DNA detection. HPV genotypes were classified as low risk (LR) and high risk (HR) by nucleotide sequencing. Results: In total, 17.3% (28/162) of the participants had HPV-DNA, and LR-HPV was the most prevalent (71.4%). Among the 13 different types of HPV detected, HPV-11 was found most frequently (12/28; 42.9%), followed by HPV-31 (3/28; 10.7%). Of the participants with cytological alterations, HPV infection was detected in only four: two were diagnosed with low-grade squamous intraepithelial lesions (15.4%), one with atypical squamous cells of undetermined significance (7.7%), and one with atypical squamous cells, cannot exclude high-grade squamous intraepithelial lesions (7.7%). Of the 61 women who presented a normal cytology, 13 (21.3%) had positive tests for HPV infection, 4 (8.2%) of which were positive for a high-risk genotype. Conclusion: The prevalence of HPV was high in Belém, Pará, and especially in women who had normal cytology results, which suggests the need for greater screening for HPV infection in women’s primary health care.

## 1. Introduction

The Centers for Disease Control and Prevention (CDC) and the World Health Organization (WHO) estimate that at least 79 million Americans are infected with human papillomavirus (HPV) [1,2,3], most in their late teens and early 20s, and about 14 million people become newly infected each year [1]. In Brazil, the incidence of cervical cancer was estimated to be 16,370 for the year 2018, with an incidence of 17.11 per 100,000 women, and it is the most frequent cancer in women from northern Brazil, with a gross incidence rate of up to 40/100.000 in some capitals [4,5].

Cervical cancer screening programs aim to detect and submit to treatment patients with precancerous lesions, avoiding the fact that these lesions may progress to cervical cancer. In the middle of the 1980s, the first national program for cervical cancer prevention (PAISM) was introduced in Brazil, and it included cervical cytology as a routine exam in gynecologic assistance [6,7,8]. However, cytology has its limitations, and even in countries with adequate cytology quality-assurance systems, these limitations still have a great influence on the performance of the screening programs [9,10,11]. 

Today, over 200 HPV genotypes are found to infect the female genital tract and are associated with tissue disorders, such as intraepithelial cervical neoplasia and cervical carcinoma. HPV genotypes are divided into two groups: low risk (LR) and high risk (HR). LR-HPV includes the genotypes 6, 11, 40, 42, 43, 44, 53, 54, 61, 72, 73, and 81, whereas HR-HPV includes the genotypes 16, 18, 31, 33, 35, 39, 45, 51, 52, 56, 58, 59, and 68. The prevalence and distribution of HPV genotypes differ according to the study population, and HR-HPV is the major recognized risk factor for the development of cervical cancer [12,13].

The HPV genome consists of a double strand of circular DNA divided into three segments. In the early segment, we found *E1*, *E2*, *E4*, and *E5,* and the oncogenic proteins *E6* and *E7*. The late segment contains the *L1* and *L2* genes that encode the proteins that are, respectively, the major and secondary capsid proteins, and finally a regulatory region (*LCR*), where the replication site is located. Some studies on HPV prevalence were performed in Brazil on a population of HIV-seronegative women who were spontaneously seeking gynecological care and prevention. The prevalence varies from 15.7% to 60% in the south [14,15,16,17], from 48.6% to 59% in the southeast [16], and in the northeast, the variation range is from 11.7% to 80.4% in communities of black rural descendants of enslaved Africans, designated as Quilombolas [18,19,20,21,22,23,24]. In the northern region, the prevalence varies from 16.4% in communities of riversides [25,26,27], to 29.4% in programs of cervical routine assistance in the largest cities.

Investigations of HPV infection and cellular abnormalities have been detected frequently, as well as the presence of some oncogenic genotypes in normal cytology samples; these genotypes are usually HPV 16, 18, 52, 58, 31, 51, and 56 [12]. Considering that, globally, almost 70% of all cervical cancers are associated with HPV 16 and/or HPV 18, the first vaccine generation included protection for both HPV genotypes [13]. Epidemiological studies that use prevalence investigations are the best way to collect and adopt data to promote public health policies, with prevention being the focus of the improved quality to assist populations living in large and small cities, or in isolated communities. The present study aims to estimate the prevalence and diversity of HPV genotypes, and to identify the related risk factors among women screened for cervical cancer in Belém, Pará, northern Brazil.

## 2. Results

### 2.1. Sample Characteristics

Cervical smear tests were carried out in 162 women for histopathological diagnosis and to detect HPV-DNA. The mean age of the population sample was 37.5 years (range: 17–76 years), 49.4% were married, and 43.8% were in high school. Most of the participants (51.8%) first had sexual intercourse between 18 and 22 years old, 58.6% never used condoms in sexual relations, 38.4% used oral contraceptives, and 82.7% had had up to five sexual partners in a lifetime (Table 1). The demographic and behavioral characteristics of the women infected with HPV were found to be similar to those of women without detectable HPV infection in terms of age, schooling level, marital status, sexually active life in the last 12 months, oral-contraceptive use, sexual partners, and previous participation in cervical cancer screening. The age of sexual initiation was the main variable associated with exposure to HPV (*p* < 0.04) (Table 1).

### 2.2. Cytological Analysis

A total of 65.4% (101/162) of the women showed cervical abnormalities, of which 61.4% (62/101) had vaginal microorganisms, and 24.7% showed inflammatory cytology. Atypical squamous cells were detected in 13 participants (8.0%), of which 2 (1.2%) represented low-grade squamous intraepithelial lesions (LSILs), 3 (1.8%) represented high-grade squamous intraepithelial lesions (HSILs), 5 (3.1%) were diagnosed as atypical squamous cells of undetermined significance (ASCUS), and 3 (1.8%) were diagnosed as atypical squamous cells, cannot exclude high-grade squamous intraepithelial lesions (ASC-H). One woman (0.6%) was diagnosed with invasive cervical cancer (Table 2). Out of the 13 women with cytological alterations, 8 (61.5%) were married, 6 (46.1%) first had sexual intercourse at an age ranging from 18 to 22 years old, 9 (69.2%) were related to the use of oral contraceptives, 7 (53.8%) never used condoms in sexual relations, and 5 (38.5%) had never submitted to a Pap-smear routine.

### 2.3. HPV Identification

HPV-DNA detection was found in 28/162 samples (17.3%), with most of them (20/28; 71.4%) representing LR-HPV. A total of 13 different HPV genotypes were detected: LR-HPV (HPV-11, HPV-61, HPV-81, HPV-6, HPV-54, HPV-72, and HPV-89) and HR-HPV (HPV-31, HPV-52, HPV-16, HPV-18, HPV-53, and HPV-58). HPV-11 was the most prevalent (42.9%), followed by HPV-31 (10.7%) (Table 2).

Of the 13 samples with cytological alterations, HPV infection was detected in only 4:2 were diagnosed with LSIL (15.4%), 1 with ASCUS (7.7%), and 1 with ASC-H (7.7%). Of the 61 women who presented a normal cytology, 13 (21.3%) had positive tests for HPV infection, 4 (8.2%) of which were positive for a high-risk genotype.

## 3. Discussion

Considering the absolute values in our study, even with the reduction in the HPV-infection incidence among elderly individuals, Brazil’s National Institute of Cancer (INCA) demonstrated that, in the north and northeast regions, there has been a slight increase in the incidence of cervical cancer since 2008 [28].

As related by some authors, cervical cancer prevention is primarily conducted by HPV-infection avoidance [29]. In our population sample, this strategy is failing, as is evidenced by the percentage of sexually active women who did not use condoms in the last twelve months (58.6% (95/162)), and the answers of women that said yes to the use of condoms could be contested by observing the number of positive samples in this group (23.8% (16/67)), which is greater than the group that does not use them (14.7%) (14/95) [30].

Cervical cytology has been effective at detecting and preventing most precancer lesions over the past sixteen years, and mainly after the introduction of the screening Papanicolaou test in the 1950s, when a large reduction in deaths by cervical cancer was observed in the United States and Canada, passing from the most common cancer death in women to the 15th cause of cancer deaths. A reduction in mortality from 50 to 80% was also observed in Europe [31].

Maybe due to the prevalence of the low-risk HPV-genotype infections associated with the profile of the spontaneous demand in our population study group, the results exhibit a relatively low incidence of high-grade lesions and/or cancer. The general positivity agrees with what has already been shown in a meta-analysis that evaluated 4354 women in Latin America, including studies held in Brazil, Argentina, Chile, Paraguay, Colombia, and Peru, where the positivity was 14.3% and could be comparable to our result of 18.2% [11,32].

In our population sample, the percentile of women that had never had a Pap smear before participating in our study was only 16.6%, and diverging from other studies with greater sample numbers, it demonstrated that only 14% of women had had a previous Pap smear. Considering that both studies were held with low-income populations, our results did not reflect the real regional cancer incidence, which is probably due to the lack of an organized and broader program for cervical cancer prevention in Brazil. Another important observation is the diversity of HPV genotypes that were found in the study population: 13 genotypes in only 30 positive samples [10,15].

Although it is the fourth most diagnosed cancer, and the fourth leading cause of cancer death, cervical cancer is almost completely preventable due to the available vaccination and screening [33,34]. In Brazil, cervical cancer is the third most common type of cancer, and the most likely among women in the north of the country [35]. This Brazilian region, which includes most of the Amazon rainforest, is a rural area, with levels of poverty, limited transport infrastructure, and inadequate services, such as low HPV-vaccine coverage in the states of Amazonas, Roraima, and Pará [36,37,38]. Thus, this study is an important record of the epidemiological scenario of HPV infection in women in northern Brazil in 2007 and 2008: 17.3% of women tested positive for cervical HPV, and there was a predominance of positive HPV genotypes (73.3%) and cytological abnormalities, such as inflamed changes and the presence of atypical squamous cells. After the period of the execution of this study, measures for uterine screening and uterine prevention were undertaken in Brazil and other countries [39]. For example, by the end of 2016, 12 countries and 1 territory (23/13, 6%) in Latin America had 5 HPV vaccine programs in their immunization countries [40].

However, increasing the high protection against HPV, and strengthening the monitoring, evaluation, and reporting of increased cancer cases and still-difficult cancer vaccines, are challenges to be overcome in several countries, which have structures of services mainly in immune health, and areas of difficult access to protect against HPV [40,41,42]. In Brazil, the prevalence of cervical HPV was 25.41% (95% CI: from 22.71 to 28.32), highlighting the profound lack of data in many Brazilian geographic areas [43]. In Argentina, there is a record of an even deeper prevalence of cervical HPV (48.2%) [42]. In Chile, the prevalence of cervical HPV varies from 11.1% to 80.8% [43,44]. High rates of cervical HPV have also been reported in other countries.

The prevalence of HPV among women in Kazakhstan (39.0%) and Egypt (40.8%) was higher than that recorded here, with a predominance of high-risk HPV genotypes, such as HPV16 and HPV18 [45,46]. A high rate of high-risk HPV (52.0%) was recorded among young women in France, and an association was observed with the degree of cytological abnormalities. However, the presence of HPV16 and HPV18, both targeted by vaccines, was extremely rare among vaccinated young French women [47]. Similarly, infections with the HPV genotypes included in the vaccine (HPV 16 and HPV18) were rare in Canadian women younger than 23 years, and they were virtually absent in those who had received at least one dose of the vaccine before sexual initiation [48]. These findings demonstrate that HPV and cervical-cancer-screening and -prevention strategies and actions can be effective, but they must be carefully planned, executed, and adjusted in underdeveloped areas, and especially considering the adversities and peculiarities that exist in a geographic region.

This is very evident in the Brazilian state of Pará, which has areas with medium HDIs, such as the metropolitan region of Belém, and other remote areas with low and very low HDIs, such as the Marajó Archipelago, southeast and southwest of the state. In recent studies, the prevalence of HPV remains similar to, or even higher, than the value recorded here (2007–2008), and the epidemiological profile of women infected with this virus is still very worrying. In 2017, 15.5% of a sample of women in the city of Belém had cervical HPV, and most of them were young, single, with an early onset of sexual life, had several sexual partners in their lifetime, and used oral contraceptives [49]. In this city, a high rate of HPV (63.3%) was also recorded among women with HIV, with a predominance of high-risk HPV genotypes, such as HPV16 (37.5%), and the detection of premalignant alterations in the samples (LSIL (39.4%), HSIL (45.4%), and ASCUS (12.1%)) [27]. In southeastern Pará, a prevalence of HPV (36.6%) was recorded among women who attended public and private health services in the city of Jacundá in 2015 and 2016, and this rate was also found among women aged 45 years or older (40.3%). In this study, 28.9% of the participants showed changes in cervical cytology, such as inflammation and the presence of atypical squamous cells of significance in cervical intraepithelial neoplasia. A low abusive use of alcohol and altered cytology results were factors associated with HPV infection [50].

These findings show that, both in the past (data from this study) and in the few recent studies, there is still a clear need for an effective program for the diagnosis of HPV and the monitoring and treatment of possible cases of cervical cancer, as well as actions for prevention and health promotion in the state of Pará. The implementation of an effective and regular program for cervical HPV diagnosis (followed by the monitoring and clinical treatment of people with cytological abnormalities) is a very promising tool for reducing cervical cancer morbidity and mortality [34]. According to Rocha et al. [51], cervical quality assurance in the female population needs to be increased, and health promotion through intersectoral cancer partnerships, popular participation, and collective accountability for quality needs to be ensured. The costs associated with lives and reducing suffering due to disease morbidity should be pragmatically addressed by governments in low-developed areas, such as Brazil, and improving outcomes should be a key priority for those who are responsible for addressing systems of cervical cancer screening.

This study has limitations that should be considered. The restriction of the study to women who sought the health service only at a public care unit in the state of Pará was the first limiting factor, indicating that the sample may not represent the entire population. Second, adolescent women were excluded by the age limit of 18 years or older. Third, cervical cytology is performed by examination and by collecting samples of cervical cells, which are either smeared onto a slide for conventional cytology or placed into a liquid medium for liquid-based cytology. Cervical cytology is a specific test and could reveal normal cells, low-grade abnormalities, or high-grade abnormalities, and in these cases, the likelihood of precancer is high. However, cervical cytology is not a sensitive test and, according to Eun and Perkins [52], from 30% to 50% of precancer diagnoses are missed with each screening round, which could be dangerous to the individual. We know that, nowadays, cervical cytology is by far not the gold-standard test for HPV due to the reason cited above; even so, here in Brazil, cervical cytology is a low-cost test and can be performed on national territory because many cities in Brazil have low-income statuses and a low-cost test might help to prevent cancer when it is frequently repeated. Finally, the ability to establish causality was limited in this cross-sectional study.

Even today, Brazil is a developing country, and within our national territory, there are many discrepancies among regions. Especially when comparing the northern and southern regions, the economic, health, and social discrepancies emerge. Some national programs were created to decrease these discrepancies, and regarding HPV diagnosis, cervical screening, and HPV vaccination, a national program was created by the Brazil Ministry of Health and is still used in the roll of the national territory to prevent as many cases as possible.

Lastly, our results corroborate some substantial data, such as the age of sexual initiation and the number of lifetime sexual partners, as important risk factors for acquiring HPV infection, and therefore, an increased risk of the development of cervical cancer. However, it is of great importance, as a strategy to reduce the cervical cancer incidence in northern Brazil, to implement a national data bank of information on routine exams in the general population [27].

## 4. Materials and Methods

### 4.1. Study Population and Ethics Aspects

This cross-sectional study was conducted between January of 2007 and December of 2008 in Belém, the capital of Pará State, in northern Brazil, and it included all women (*n* = 162) who were spontaneously undergoing a Pap-smear routine (cervical cancer screening program) at the Cytopathology Laboratory in the Biological Science Institute of the Federal University of Pará. The study excluded pregnant women, virgins, women aged ≤ 18 years, and patients with a history of HPV infection or cervical cancer.

The Ethics and Research Committee of the Health of Science Institute approved the study under protocol number 805/2006, and written informed consent was obtained from all participants prior to enrolment.

### 4.2. Study Design and Procedures

A structured questionnaire was used to collect information about the socioeconomic and epidemiological characteristics, as well as age, education, sexual history and behavior, condom usage, sexual partners over the lifetime, marital status, and smoking history.

Members of the study group performed the interviews with participants and collected cervical samples (endocervix and ectocervix) for cytological examination and for HPV-DNA detection. For cytological analysis (Pap smear), two cervix samples were collected with an Ayre spatula and were spread on glass slides, immediately fixed in alcohol (96%), stained by the Papanicolaou method, and examined by an expert cytologist. The results from the cytological tests were classified according to the Brazilian guidelines for uterine cervical cancer screening in LSIL, HSIL, ASCUS, and ASC-H. For the HPV-DNA detection, cervical samples were obtained using a cytological endocervical brush, transferred to a sterile polyethylene tube containing 2 mL of saline solution, and stored at −20 °C until DNA extraction [53].

### 4.3. DNA Extraction and HPV Detection

Genomic DNA was extracted using the phenol–chloroform standard method, and the control for the DNA quality was evaluated by the amplification of the human *MBL* gene in all samples [54,55].

HPV-DNA amplification was performed by polymerase chain reaction (PCR) using degenerate primers (MY09 (CGTCCMARRGGAWACTGATC) and MY11 (GCMCAGGGWCATAAYAATGG)), which amplified a 450 bp fragment of the HPV *L1* gene [56,57]. Briefly, the amplification reaction was carried out in a 50 μL reaction volume by using 200 ng of DNA, 10 mM of each dNTP, 30 μM of each primer, 1.5 mM of MgCl2, 1X PCR buffer (10 mM Tris-HCl; pH: 8.3; 50 mM KCl), and 0.5 μL (1 unit) of *Platinum Taq* DNA polymerase (Invitrogen^TM^, Waltham, MA, USA). Amplification conditions consisted of the following protocol: 40 cycles at 95 °C for 5 min, followed by 56 °C for 40 s, and 1 min at 72 °C. A final extension of 72 °C was performed for 5 min. Amplification products were submitted to 2% agarose gel in 1X TAE buffer with Syber-Green (Invitrogen^TM^, Waltham, MA, USA) for 30 min, and were visualized under ultraviolet light. The samples that were positive in the PCR were submitted to DNA sequencing (in duplicate), in both directions, by using ABI PRISMTM 310 BigDye Terminator v 3.1 Matrix Standards (Applied Biosystems, Foster City, CA, USA), with the ABI 3130 genetic analyzer (Applied Biosystems, Foster City, CA, USA), as described by the manufacturer. Nucleotide sequences were edited using BIOEDIT software, version 5.0.9 (Los Angeles, CA, USA), and sequence comparisons were performed using the Web BLAST Tool Nucleotide Blast (https://blast.ncbi.nlm.nih.gov/Blast.cgi, accessed on 31 January 2022) to identify the HPV genotype [28].

### 4.4. Statistical Analysis

The epidemiological and demographic data were entered into a database using Access 2010 software (Albuquerque, NM, USA). The HPV genotypes were classified as high risk (HR) and low risk (LR), according to their oncogenic potential. Associations between epidemiological and demographic factors and HPV infection were determined with a chi-square test and G test. We considered *p* < 0.05 to be statistically significant.

## 5. Conclusions

The high prevalence of HPV and genotypes with high oncogenic potential clearly indicates a risk to the health of women in the state of Pará, both in the past and today. Despite the record of reduction in the frequencies of the HPV16 and HPV18 genotypes among women in the city of Belém, it is still urgent to implement a more effective program for the diagnosis of HPV and the monitoring and treatment of women with cytological alterations of the cervix, and especially with cervical cancer, as well as regular prevention and health-promotion actions for the population of the city of Belém, and other cities in northern Brazil, adapted according to local peculiarities and adversities.

## Figures and Tables

**Table 1 pathogens-11-00960-t001:** Demographic and epidemiological characteristics of the 162 women screened for cervical cancer in Belém, Pará, from January 2007 to December 2008, considering the HPV positivity results.

Characteristics		HPV Infection		
	Total	Positive (%)	Negative (%)	*p*-Value ^a^
Age (years)				
17–28	49	10 (20.4)	39 (79.6)	0.17
29–40	52	13 (25.0)	39 (75.0)	
>40	61	7 (11.5)	54 (88.5)	
Educational Level				
Incomplete elementary school (including illiterate) ^b^	26	3 (11.5)	23 (88.5)	0.48
Complete elementary school	13	2 (15.4)	11 (84.6)	
High school (complete + incomplete)	71	12 (16.9)	59 (83.1)	
University (complete + incomplete)	52	13 (25.0)	39 (75.0)	
Age of sexual initiation (years)				
13–17	61	12 (19.7)	49 (80.3)	0.04
18–22	84	11 (13.1)	73 (86.9)	
>22	17	7 (41.2)	10 (58.8)	
Marital status				
Single, divorced, or widow	82	19 (23.2)	63 (76.8)	0.11
Married or cohabitating	80	11 (13.8)	69 (86.2)	
Sexually active life in the last 12 months				
Yes	137	24 (17.5)	113 (82.5)	0.44
No	25	6 (24.0)	19 (76.0)	
Condom use in sexual intercourse in the last 12 months				
Yes	67	16 (23.9)	51 (76.1)	0.14
No	95	14 (14.7)	81 (85.3)	
Oral contraceptive use in the last 12 months				
Yes	62	11 (17.7)	51 (82.3)	0.84
No	100	19 (19.0)	81 (81.0)	
Number of different sexual partners over a lifetime				
0–2	87	17 (19.5)	70 (80.5)	0.82
3–5	47	9 (19.1)	38 (80.9)	
>5	28	4 (14.3)	24 (85.7)	
Previous participation in cervical cancer screening				
Yes	135	17 (12.6)	118 (87.4)	0.41
No	27	5 (18.5)	22 (81.5)	

^a^ Calculated by chi-square or G tests. ^b^ Less than eight years of study.

**Table 2 pathogens-11-00960-t002:** Prevalence of cytological abnormalities and HPV infection in women screened for cervical cancer in Belém, Pará, northern Brazil, from January 2007 to December 2008.

Diagnosis	% (Positive/Total)	95% Confidence Interval
Cytological changes		
Cervical abnormalities	62.3 (101/162)	59.2–66.2
Some inflammatory cytology	24.7 (40/162)	19.2–30.7
Atypical squamous cells	8.0 (13/162)	2.8–12.3
LSIL	1.2 (2/162)	0.0–4.3
HSIL	1.8 (3/162)	0.0–4.8
ASCUS	3.1 (5/162)	0.0–6.2
ASC-H	1.8 (3/162)	0.0–4.8
Invasive cervical cancer	0.6 (1/162)	0.0–4.1
HPV infection		
HPV-DNA	17.3 (28/162)	14.6–22.9
Low-risk (LR) genotypes	71.4 (20/28)	69.9–78.1
Genotype 11	42.9 (12/28)	35.8–45.3
Genotype 61	7.1 (2/28)	1.9–10.8
Genotype 81	7.1 (2/28)	1.9–10.8
Genotype 6	3.6 (1/28)	0.0–7.9
Genotype 54	3.6 (1/28)	0.0–7.9
Genotype 72	3.6 (1/28)	0.0–7.9
Genotype 89	3.6 (1/28)	0.0–7.9
High-risk (HR) genotypes	28.6 (8/28)	22.4–32.5
Genotype 31	10.7 (3/28)	5.9–13.6
Genotype 52	3.6 (1/28)	0.0–7.9
Genotype 16	3.6 (1/28)	0.0–7.9
Genotype 18	3.6 (1/28)	0.0–7.9
Genotype 53	3.6 (1/28)	0.0–7.9
Genotype 58	3.6 (1/28)	0.0–7.9

## Data Availability

All data referred to in this study are available in the manuscript.

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
