# Peer review of "Prevalence, Diversity, and Risk Factors for Cervical HPV Infection in Women Screened for Cervical Cancer in Belém, Pará, Northern Brazil"

_pathogens, 2022, doi:10.3390/pathogens11090960_

Round 1
Reviewer 1 Report
The manuscript title and the abstract clearly present the text. However, results presented in the abstract (lines 32-37) sound unclear and difficult to comprehend, therefore, suggested being rewritten.
The introduction part gives a clear rationale of the study.
Materials and methods cover all necessary information.
Results are clear, however, could be presented in a better way by including figures to present the distribution of the different HPV types identified among the study participants.
The discussion part is interesting. However, as one of the most important parts of the manuscript, it should be improved by minor restructuring.
The following structure of the discussion part is suggested:
1.1 Rationale of the study (why it was done)
1.1.1 Main findings of the study
1.1.2 What makes our study unique
1.1.3 What it adds to what we already know
1.2 Study subjects
1.3 Subject of the discussion
Comparison of our results with neighboring countries, with countries of the same development levels (income), with developed high-income countries). Agreement and disagreement with the studies compared. Suggested recent studies on the same topic to compare (doi: 10.1177/0969141320902482; doi: 10.1016/j.ijid.2021.06.006; doi: 10.1177/17455065211004135;doi: 10.3390/biology10080794; doi: 10.1016/j.ijid.2016.11.410).
1.4 Sum up of the study, study strength and limitations
1.5 Clinical implication of the stusy findings
2The manuscript conclusion should be improved to represent the study findings in a better way.
Author Response
Dear Reviewer 1
We woulda like to thank you for the manuscript review and all your revisions, the answers to all these revisions are highlighted in yellow
Reviewer 2 Report
This submission describes a rather small (162 pts) cross-sectional study conducted several years ago (2007-8). During these 15 years the understanding of HPV-related disease has transformed and guidelines on cervical screening policies have vastly evolved globally. While this study might have been perfectly adequate fifteen years ago, unfortunately now it lacks substantial novelty, especially in the context of a special issue focusing on “Emerging and Re-emerging Viral Infectious Diseases”.
Major Points:
1. No data are provided on cervical screening infrastructures in Brazil. No data are quoted on primary cervical prevention schemes within the main Brazilian territories (North/Northeast/Southeast/South/Central west) and the corresponding HPV vaccination coverages.
2. The Discussion section is poorly developed.
Minor points:
1. Suboptimal use of English, sometimes causing difficulties in the comprehension of the text (e.g. the first paragraph of the Discussion, rows 141-145).
2. Under the current understanding, cytology is not indicated for individuals aged less than 18years (row 176).
Author Response
Dear Reviewer 1
We woulda like to thank you for the manuscript review and all your revisions, the answers to all these revisions are highlighted in green
Round 2
Reviewer 2 Report
In this revised version the manuscript was significantly enriched and improved (except the marked novelty of mixing Portuguese words/terms in the English text). Hence, it might now be eligible for publication in “Diagnostics”, providing further changes are made throughout the entire text to significantly improve language and cohesion.
Comparisons with Kazakhstan (row 323) and Egypt (row 324) are unfortunate because of the geographic and social diversity.
A major drawback of this study remains its conduction in the pre-HPV vaccination era (2007-8), the presented data could be considered outdated now.
Author Response
Several changes have been made to this new version of the manuscript " Prevalence, diversity and risk factors for cervical HPV infection in women screened for cervical cancer in Belém, Pará, Northern Brazil" (pathogens-1825969), which have been highlighted in green.
Responses to evaluator 2's comments were made (below), some of which are highlighted in this new version of the manuscript.
The authors thank the reviewers for their attention and comments and await the editorial decision.
Sincerely,
Prof. Dr. Luiz Fernando Almeida Machado.
Responses to comments - reviewer 2:
- The manuscript has been completely revised for language corrections.
- The authors expanded the comparison of the findings of this study with the results of other countries (Kazakhstan, Egypt, France, and Canada), and evidenced the relationship that strategies and actions for the prevention and screening of HPV and cervical cancer can be effective, but they must be carefully planned, executed, and adjusted in underdeveloped areas, especially considering the adversities and peculiarities existing in a geographic region as in the immense state of Pará. Furthermore, the authors highlight here that the inclusion of epidemiological data from Kazakhstan was a request from the other reviewer.
- The authors disagree with the position of reviewer 2 regarding the disadvantage of this study having been conducted in the pre-vaccination period. The findings of this study compared to the few studies conducted in the last decade (vaccination period) clearly indicate that the control and prevention program needs adjustments, as there is still a high frequency of high-risk HPV genotypes being detected among the women, particularly in geographic areas with inadequate health services. This comparison (past versus present) can be safely made because of the findings of this study. This position was highlighted in the discussion and conclusion of the study.